# Whence the 8th Day of the 4th Lunar Month as the Buddha's Birthday

Meiqiao Zhang 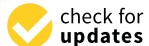

School of History, Zhejiang University, Hangzhou 310058, China; meiqiaozhang@outlook.com or meiqiao@zju.edu.cn

**Abstract:** Two dates, the 8th day of the 4th lunar month (Date A) and the 8th day of the 2nd lunar month (Date B), are found in Chinese Buddhist translations as the Buddha's birthday. However, how to understand the simultaneous existence of both of these dates remains an unresolved problem. This paper proposes a rather new interpretation to try to solve this puzzle, and provide an answer to the question: whence the 8th day of the 4th lunar month as the Buddha's birthday? It is argued that: (1) The date of the Buddha's conception and the date of his birth were both translated variously as Date A or Date B in early Chinese Buddhist literature. However, many later texts referring to the Buddha's birthday do not include reference to an auspicious junction star (*puṣyanakṣatra*), which is critical for understanding these dates; (2) Both the Indian and Chinese traditions regard an individual's life to begin at the moment of conception; therefore, the so-called Buddha's birthday could be argued as the date of his conception; (3) The date of conception of the Buddha was specified as the 8th day of the *śuklapakṣa* of the month Vaiśākha, the day of the vernal equinox. This corresponds to Date A in the Chinese Xia calendar.

**Keywords:** Buddha's birthday; *puṣyanakṣatra*; April 8; Vaiśākha; Sino-Indian calendar



## 1. Introduction

Two dates, the 8th day of the 4th lunar month (Date A) and the 8th day of the 2nd lunar month (Date B), are found in Chinese Buddhist translations as the Buddha's birthday. However, how to understand the simultaneous existence of both of these dates remains unresolved. As is well known, much controversy has surrounded the question of the dates of the Buddha, and in particular the year of his death, which was the main topic of an international conference organized by Heinz Bechert at Göttingen in 1988. Although the date of the Buddha's birth was also touched on (Bechert 1991–1997, p. 97), little was settled; later, other conferences also dealt with the Buddha's birth (Cueppers et al. 2010), but the topic, and in particular the question of how to account for the inconsistent dates in the Chinese literature, has largely been overlooked compared to the attention paid to the date of the Buddha's death. Date A has been tacitly recognized as the appropriate time for celebration of the Buddha's birthday in most of East Asia.[1] However, in regard to the inconsistent dates for the Buddha's birth and the reasoning for taking either as the Buddha's birthday, there is still a lack of scholarly consensus.

Noteworthy is the publication of *Shakuson no shōgai ni sotte hairetsu shita jisekibetsu genshi bukkyō seiten sōran* 釈尊の生涯にそって配列した事績別原始仏教聖典総覧, by Mori Shōji 森章司 and his research team (Shakusonden Kenkyūkai 1999–2019). Mori distinguished between the date of the Buddha's conception and that of his birth (Mori 1999). Specifically, he distinguishes "the date the Buddha descended into the womb 入胎日" (the date of conception) from "the date the Buddha came out of the womb 出胎日" (the date of birth) respectively, according to Chinese translations of the Buddha's biographical literature. However, which of the two events corresponds to Date A and which to Date

B remains to be established. This inspired me to consider the unresolved confusion surrounding the two dates found for the Buddha's birthday.

In this essay, I start from materials related to a misunderstood term in Chinese literature, both Buddhist and non-Buddhist, and then look back to possible sources and their Indian Buddhist background. I not only rely on counterparts in different languages but also pay attention to information supplied by Indian and Chinese astronomy. Based on these analyses, I argue that a date combining month (月) and day (日) did not suffice for correctly transmitting the Buddha's date of birth; rather, we must take into account extra details, including reference to an auspicious junction star. In so doing, we can conclude that Date A, which has traditionally been taken as the Buddha's birthday, in fact pertains not to the date of birth but to the date of conception of the Buddha.

## 2. An Indispensable Element in the Moment of the Buddha's Birth: An Auspicious Junction Star

Date A is not only acknowledged in the Chinese biographical literature of the Buddha (see below), but can also be found in some non-Buddhist Chinese literature. For example, as Erik Zürcher (2007, pp. 271–72) notes, Chinese non-Buddhist literature used two systems for dating the Buddha's birth, arriving at two different dates: the first on Date A of the 10th year of King Zhuang of Zhou 周庄王 (now identified as a period ending in 682 BCE, but whose beginning is unknown), the other on Date A of the 24th year of King Zhao of Zhou 周昭王 (now identified as ?1027 BCE–977 BCE). Ancient scholars usually relied on the latter date to argue for Buddhism's superiority over Daoism, and these references are the reason that these dates are frequently quoted even by modern scholars (Liu 2017; Chen 2018). What is more, Zürcher (2007, p. 272) suggests that one of the Chinese translations of the Buddha's biography, the *Taizi ruiying benqi jing* 太子瑞應本起經, "says that 'on the 8th day of the 4th month, *when the night was bright* (夜明) the Buddha was born', using exactly the words of the *Zuozhuan* [左傳] passage mentioned" (emphasis in original), and most scholars have adopted Zürcher's argument. However, it is well worth our while to reexamine the contents of the *Zuozhuan* and *Taizi ruiying benqi jing*.

The *Zuozhuan* is a commentary on the *Spring and Autumn Annals* (*Chunqiu* 春秋), and regarding the 7th year of King Zhuang of Lu 魯莊公（now identified as 706 BCE–662 BCE) it says, "in the summer, a star could not be seen [because] the night was bright".[2] As the two Chinese characters "night 夜" and "bright 明" are also found in the later *Taizi ruiying benqi jing*, Zürcher may have thought that the latter borrowed exactly these two words from the former. Zürcher read "夜明" as "the night was bright" and assumed it refers to Date A. However, the whole content of this sentence in the *Taizi ruiying benqi jing* should rather be translated, "when it was the night of the 8th day of the 4th lunar month, *with the appearance of the bright star* (明星出時) (emphasis added), [the Bodhisattva] came out from [Māyā's] right side".[3] Therefore, it is clear that "night 夜" and "bright 明" are separate from each other, as reflected in my insertion of a comma between them (see note 3), instead of comprising one single term being a supplement for Date A, as understood by Zürcher. Thus, the meaning of the "bright star 明星" and how it influences the record of the Buddha's birth needs to be reconsidered carefully.

### 2.1. Star Fei 沸星

The earliest accounts of the Buddha's birth which include both the date and the name of a star are Chinese translations of the *Mahāparinibbāṇa Sutta*. However, it is worth noting that there is no corresponding section in the Pali *Mahāparinibbāṇa Sutta* of the *Dīgha nikāya*.[4]

The *Youxing jing* 游行經 (one Chinese parallel of the *Mahāparinibbāṇa Sutta*) in the *Chang ahan jing* 長阿含經 (the Chinese parallel of the *Dīgha nikāya*), attributed to Buddhayaśas 佛陀耶舍 (date unknown) and Zhu Fonian 竺佛念 (4–5c), states:

> How was [the Buddha] born as the one supremely honored among the two-legged (human beings)? How did [he] become free from transmigration? How did [he] attain the supreme way? How did [he] enter into the citadel of *nirvāṇa*?

It was under the star Fei [that the Buddha] was born as the one supremely honored among the two-legged … … It was on the 8th day that the Tathāgata was born … … It was on the 8th day [that the Buddha was] born as the one supremely honored among the two-legged … … It was in the 2nd lunar month that the Tathāgata was born … … It was in the 2nd lunar month [that the Buddha] was born as the one supremely honored among the two-legged … … [5]

In this section, the star Fei 沸星 is a necessary element comprising the time of the same four events in the Buddha's life (birth; freedom from transmigration; attainment of the supreme way; entering into *nirvāṇa*) together with two expressions: on the 8th day 八日, and, of the 2nd lunar month 二月. A similar structure with star Fei, date and month is found in two other Chinese parallels of the *Mahāparinibbāna Sutta*, but they differ from the translation above in that they render the 8th day of the 4th lunar month (Date A) instead of the 2nd lunar month (Date B).[6] I will come back to this discrepancy in the two dates later; in the following section, I demonstrate that the bright star 明星 as introduced above can be identified with the star Fei as described in the three Chinese translations of the *Mahāparinibbāna Sutta*.

### 2.2. Star Fei's Other Names or Interpretations: Bright Star 明星, Fusha 弗沙, and Gui Mansion 鬼宿

A 12th century glossary of Chinese Buddhist translation terms, the *Fanyi mingyi ji* 翻譯名義集, edited by Fayun 法雲 (1087–1158), explains the star Fusha 弗沙 and its several different names, including star Fei, in the following way:

Fusha 弗沙 is correctly named Fusha 富沙. [Master] Qingliang says: "It is also named 'Bo sha 勃沙'. Here[7], it [Fusha 弗沙] means 'prosper' because of the meaning of insight and ultimate truth".[8] [Fusha 弗沙] is also named Tiṣya 底沙,[9] Tishe 提舍 as well. Here, it is translated as 'bright' 明. In addition, it is named Shuodu 説度 [namely], preach the Dharma and liberate others. [Separately,] Master [Kumāra]jīva interprets the [name of] Fusha Bodhisattva: "[Star Fusha is] the name of the star Gui 鬼, [which is also the name of one of] the Twenty-Eight Mansions. When [someone is] born under the Guixiu 鬼宿, they are named after it. Alternatively, [they can be] named after the star Fei, or the star Bei 孛星." (Emphasis added.) [10]

From the above it is clear that the star Fei has several different names and semantic interpretations; in addition to the translation "bright 明", the most commonly found names in Chinese Buddhist literature related to the Buddha's birth are Fusha 弗沙 and Guixiu 鬼宿. To take these three terms in turn, let us begin with "bright". Fayun equates Fusha with the quality bright, while Kumārajīva states that Fusha is synonymous with both Gui and Fei. Thus, it seems feasible to tentatively identify the star Fei with the quality "bright". This connection between Fusha and Fei supports our understanding that "the bright star 明星" is the star Fei, as mentioned above.

Secondly, regarding Fusha, we can locate it in one of the Chinese translations of the *Lalitavistara*, the *Fangguang dazhuangyan jing* 方廣大莊嚴經 (henceforth FGJ) translated by Divākara 地婆訶羅 (613–687). There, the Chinese Fusha appears to be a transliteration of the Sanskrit *puṣya*, the name of the *nakṣatra* (constellation) of the month Pauṣa: "The Bodhisattva did not enter the womb in Kṛṣṇapakṣa, but in conjunction with the star Fusha in Śuklapakṣa."[11] The relevant Sanskrit text, which does not exactly correspond to the Chinese, reads: "*na khalu punar mārṣāḥ kṛṣṇapakṣe bodhisattvo mātuḥ kukṣāv avakrāmati, api tu śuklapakṣe evaṃ pañcadaśyāṃ pūrṇāyāṃ pūrṇimāyāṃ puṣyanakṣatrayoge*" (Hokazono 1994, p. 316). We also find a very similar description about the conception of Dīpaṃkara in the *Mahāvastu*: "*pūrṇāyām pūrṇamāsyām puṣyanakṣatrayogayuktāyām*."[12] J. J. Jones (1949, p. 162) renders this passage as "on the day of the full moon in the month Pauṣa". In a footnote regarding this translation, Jones states: "Literally 'when the full moon is in conjunction with

the asterism or lunar mansion, puṣya,' *pūrṇāyāṃ pūrṇamāsyāṃ puṣyanakṣatrayogayuktāyāṃ*; whence the name of the month Pauṣa, corresponding to December–January". Jones's interpretation as "the month of Pauṣa" seems reasonable because in India a month's name follows the *nakṣatra* in which there is a conjunction with the full moon in Śuklapakṣa.[13] However, if this interpretation is correct, then there is a contradiction regarding the month of the Bodhisattva's conception: if it is during December–January,[14] then the birth would be during September–October. This is far from both Date A and Date B. I will come back to this question of the date later. Returning to the star Fusha and its relation to *puṣyanakṣatra*, K. V. Sarma (2008, p. 1242) states, "*Yogatārā* (junction star) is the cardinal star in a *nakṣatra* which is made up of several stars. Normally, the *Yogatārā* would be the brightest star in the group, and the zodiacal signs would mostly be named after that star". Thus, the star Fusha should be the junction star of *puṣyanakṣatra*.

This brings us to the third term connected to Fusha, the Guixiu 鬼宿. According to Kumārajīva, *puṣyanakṣatra* (Fusha) corresponds to the Guixiu.[15] Further, in the *Foben xingji jing* 佛本行集經, translated by Jñānagupta 闍那崛多 (523–601?), we find the following reference to Guixiu: "On the day of Guixiu 鬼宿日, the Bodhisattva descended into his mother's womb".[16] In contrast to the FGJ, the inclusion in the *Foben xingji jing* of the character 日 in the phrase 鬼宿日 provides a more specific date regarding the conjunction between star Fusha and the moon. This phrase appears to have a parallel in the passage from the *Mahāvastu* presented above, dating the conception of Dīpaṃkara to *puṣyanakṣatra*.[17] Thus, this further supports the hypothesis that the two terms, Guixiu and *puṣyanakṣatra*, are equivalents.

Pingree and Morrissey (1989, p. 99) argue that "there is no basis for identifying the stars included in the Vedic *nakṣatra*, and therefore no grounds for comparing them, for example, with the Chinese lunar mansions". However, as Needham (1974, p. 69) states, "only nine of the twenty-eight *hsiu* determinatives are identical with the corresponding *yogatārās* or 'junction stars' of the Indians, while a further eleven share the same constellation but not the same determinative star". Further, Stephenson (2008, p. 1241) affirms that "it has been shown that early Indian astronomers did not fully agree on which stars were regarded as *yogatārā*". Thus, the star Fei may well be the junction star of Guixiu (*puṣyanakṣatra*), but we cannot decide whether it would also appear as part of some other constellation. Moreover, the day of Guixiu has been dealt with as a separate topos in Chinese translations of non-biographical Buddhist literature, where it is often treated as an auspicious omen in contexts which do not discuss the date of the Buddha's birth. For example, we see, "A day with the star Fei would be an auspicious day" in the *Shisong lü* 十誦律,[18] "Guixiu is the owner of all kings or ministers",[19] and "among all of the *nakṣatras*, Guixiu is the best", according to Śubhakarasiṃha 善無畏 (637–735).[20]

What is worth noting is that this auspicious junction star has been conceived of as a symbol of the Buddha's birth in two editions of a particular Chinese Buddhist translation related to astrology entitled *Ratnaketudhāraṇīsūtra* 寶星陀羅尼經 translated in 630 by Prabhāmitra (Prabhākaramitra) 波羅頗蜜多羅 (565–633): the second edition of the Korean canon 高麗藏再雕版 and the Jin canon 金藏, which may reflect the Kaibao canon's 開寶藏 readings, attributed to the Central lineage 中原系統 of the Chinese Tripiṭaka transmission (Fang 1991, p. 246; Chikusa 1993, pp. 10–17). These two texts read: "Śākyamuni was born under the star Fusha [Fei] (which is called Guixiu in the Tang Dynasty)."[21] However, the reading "Śākyamuni 牟尼今者" is not found in the following four canons: Song Sixi canon 宋思溪藏, Yuan Puning canon 元普宁藏, Ming Jiaxing canon 明嘉興藏 (corresponding in Japan to the *Ōbakuzō* 黃檗藏), and the Song edition belonging to the Library of the Imperial Household 宮内省圖書寮本.[22] The appearance of "牟尼今者" in the former two editions implies that people in some Chinese regions knew the connection between this auspicious junction star and the Buddha's birth.

Furthermore, when we trace the auspicious junction star Fei back to early Chinese translations of the Buddha's biographical literature, we will notice that the bright star

(明星) has been taken as indicating the date of both the Buddha's conception and his birth. For example, on the one hand the *Guoqu xianzai yinguo jing* 過去現在因果經 states that "it was during the appearance of the bright star on the 8th day of the 4th lunar month (Date A) [that the Bodhisattva] descended into his mother's womb".[23] However, as cited above, according to the *Taizi ruiying benqi jing*, "when it was the night of the 8th day of the 4th lunar month (Date A), with the appearance of the bright star, [the Bodhisattva] came out from [Māyā's] right side".

So far, we have confirmed two facts: (1) we are not able to establish the precise time of the Buddha's birth based only on the month (月) and day (日). It is necessary also to incorporate the auspicious junction star, star Fei, known earlier as bright star (明星), along with the month and day; (2) the dates of conception and birth of the Buddha were translated, sometimes with the bright star, sometimes without, variously as Date A or Date B in early Chinese Buddhist literature.

It is widely accepted that as an auspicious omen, the star Fei with Date A or Date B has been taken as the same time for the four main events of the Buddha's life. However, the inconsistent Chinese renderings raise a vexing question, that is, how and why has Date A prevailed rather than Date B as a definite date of the Buddha's birthday?

### 3. The Derivation of the Buddha's Birthday: The Date of Conception Instead of Birth

In order to resolve the confusion between Date A and B, it is necessary to clarify the difference between the date of conception and the date of birth when it comes to the Buddha's birthday.

#### 3.1. Conception Month: Vaiśākha

A comprehensive survey of the following two relevant portions from the *Lalitavistara* and its two Chinese translations, *Pu yao jing* 普曜經 (henceforth PYJ) attributed to Dharmarakṣa 竺法護 (239–316), and the FGJ, reveals that a specific date is indicated for the Buddha's conception but not for his birth. Regarding conception, we read the following:

Garbhâvakrānti-parivartaḥ

*iti hi bhikṣavaḥ śiśirakālavinirgate vaiśākhamāse viśākhānakṣatrânugate ṛtupravare vasantakālasamaye . . . ṛtukālasamayaṃ pañcadaśyāṃ pūrṇamāsyāṃ poṣadha(pari)gṛhītāyā mātuḥ puṣyanakṣatrayogena bodhisattvas tuṣitavarabhavanāc cyutvā . . . jananyā dakṣiṇāyāṃ kukṣāv avakrāma[t] . . .* (Hokazono 1994, pp. 386.1–8)

Chapter on Descending into the Womb

Thus, o Bhikṣus! Winter had passed and it was the month of Vaiśākha, during the *viśākhā nakṣatrā*, the best season, at the time of the spring equinox . . . at the proper time, on the 15th day, at the full moon, by his mother observing a fast, when [the moon] was in conjunction with *puṣya nakṣatra*, the Bodhisattva left his excellent palace in Tuṣita and . . . entering into the right side of the belly of the mother . . .

PYJ: "降神處胎品"：佛語諸比丘："于時菩薩過冬盛寒，至始春之初，修合星宿，春末夏初 . . . . . . 適在時宜， 沸星應下 ，菩薩便從兜術天上，垂降威靈 . . . . . . "

(T no.0186, 491: a25–a29)

FGJ: " 處胎品": 佛告諸比丘："冬節過已， 于春分中毗舍佉月 ，叢林花葉，鮮澤可愛，不寒不熱。 氐宿合時 ，三界勝人，觀察天下，白月圓淨， 而弗沙星，正 與月合 。菩薩是時從兜率天宮沒 . . . . . . " (T no.0187, 548: c07–c10)

As for the Buddha's birth, we read the following:
Janma-parivartaḥ

*iti hi bhikṣavo daśamāseṣu nirgateṣu bodhisattvasya janmakālasamaye pratyupasthite rājñaḥ śuddhodanasyôdyāne dvātriṃśatpūrvanimittāni prādurabhūvan . . . puṣyaṃ ca nakṣatrayuktam (abhūt).* (Hokazono 1994, pp. 428.1–3, 430.2–3)

Chapter on Birth

Thus, O Bhikṣus! When ten months passed and the time of the Bodhisattva's birth had drawn near, thirty-two previous signs appeared in the garden of King Śuddhodana . . . [the moon] was in conjunction with *puṣya nakṣatra.*

PYJ: "欲生時三十二瑞品"：佛語比丘："滿十月已，菩薩臨産之時，先現瑞應三十有二 . . . . . . 沸宿下侍 . . . . . . " (T no.0186, 492: c26–493: a22)

FGJ: "誕生品"：爾時佛告諸比丘："菩薩處胎，滿足十月，將欲生時，輪檀王宮先現三十二種瑞相 . . . . . . 弗沙之星將與月合 . . . . . . " (T no.0187, 551: b29–c23)

As we can see, for the *Lalitavistara* and its two Chinese counterparts, there are several quite particular and rather definite phrases for the date of conception, while there is only one expression emphasizing ten months' pregnancy in the chapter on birth. Moreover, while in the chapter on conception star Fei (*puṣyanakṣatra*) appears as an element of the combination with other time expressions indicating a precise date, it is simply one of the thirty-two auspicious appearances in the chapter on birth.[24]

The month Vaiśākha is the first and the most vivid indication suggesting when the Buddha descended into the womb. Its counterpart cannot be found in the PYJ, but in the FGJ it is transliterated as " 毗舍佉月 Bi she qu yue" and translated as "氐宿 Dixiu". In addition, Fayun says: "Vaiśākha 毘舍佉, otherwise [may be called] 'Bi she qu' 鼻奢佉. Here [in China] it is called 'Bie zhi' 別枝, that is 'Dixiu' 氐宿".[25]

Although the month Vaiśākha only appears in the chapter on conception in the *Lalitavistara*, it would be hasty to think of it as exclusively the time of conception. According to Xuanzang (602–664), who stayed in India for approximately 17 years, all eight junctures of the Bodhisattva's life were taken as occurring in the month Vaiśākha (See Ji (2000, pp. 523, 533, 539, 678)). In the same way, all those key events in the life of the Buddha are considered either as Date A or Date B by the above three Chinese counterparts of the *Mahāparinibbāṇa Sutta*. That is, the information regarding the date of conception does have a connection with all the dates of the main events of the Buddha's life.

### 3.2. The First Step of the Buddha's Life: Conception

At first, *Jiangsheng pin* 降生品 of FGJ and *Jiangshen chutai pin* 降神處胎品 of PYJ state that Māyā had been decorating exquisitely to prepare for the Bodhisattva entering and staying in the womb, when all the gods declared that they would be in the service of the Buddha from "the beginning of his entering the womb, to his coming out of the womb, his childhood, his feeling of recreation and indulgence of desires, leaving home and practicing austerities, reaching the Bodhi tree, subduing Māra, rolling the wheel of the true Dharma, manifesting divine powers, staying in the Trayastrimśa, and entering *nirvāṇa*".[26] All of these formulations suggest that conception, rather than birth, is taken as the start of the Buddha's present life.[27]

Next, let us observe the importance of this information from the whole picture of the three stages of the Buddha's birth: the conception, the pregnancy, and the birth (see Table 1).[28]

**Table 1.** The three stages of the Buddha's birth in the *Lalitavistara* and their Chinese translations (PYJ and FGJ).

| Three Stages | Chapter of *Lalitavistara* | Chapter of PYJ | Chapter of FGJ | Chapter of Hokazono's Japanese Translation |
|---|---|---|---|---|
| 1.Conception | Pracalana-parivartaḥ | a. 所現象品＋ b. 降神處胎品 | 降生品 | 出立品 |
| 2. Pregnancy | Garbhâvakrānti-parivartaḥ | b. 降神處胎品 | 處胎品 | 入胎品 |
| 3. Birth | Janma-parivartaḥ | c. 欲生時三十二瑞品 | 誕生品 | 誕生品 |

What is noteworthy is that:

(1)     The second chapter of the PYJ (*jiangshen chutai pin* 降神處胎品) corresponds to the latter part of the first Sanskrit chapter and all of the second.

(2)     The PYJ and the FGJ translate the title of the first Sanskrit chapter differently, though they both use the Chinese character *jiang* 降.[29]

(3)     For the conception stage, FGJ adopts *jiangsheng* 降生; for the pregnancy stage, PYJ uses *jiangshen* 降神; and for the birth stage, in FGJ we find *dansheng* 誕生.

Different translations of the titles of the three chapters in which the above three stages occur lead to some difficulty clearly identifying them. For example, if we do not separate the whole birth process into three stages, both *jiangsheng* 降生 and *dansheng* 誕生 might be taken as giving birth, but the former pertains to the stage of conception while the latter has been accepted to express an individual's birthday in modern society.

In the early 20<sup>th</sup> century, the scholar Zhou Zumo 周祖謨 gives a commentary on the following entry in the 6<sup>th</sup> century text *Luoyang qielan ji* 洛陽伽藍記, a report on the monasteries of Luoyang: "A white elephant with six teeth carried Śākyamuni on its back through the void 作六牙白象，負釋迦在虛空中". Zhou's commentary is as follows[30]:

案《修行本起經》卷上云：'能仁菩薩化乘白象，來就母胎。'《普曜經》卷二云：'菩薩從兜術天上化作白象，口有六牙，降神母胎。' 此云作六牙白象，負釋迦在虛空中者，即佛降生之相也。

According to the *Xiuxing benqi jing*: 'Bodhisattva Nengren transformed into a white elephant and entered [his] mother's womb.' According to the second chapter of the PYJ 'The Bodhisattva transformed into a white elephant with six teeth and descended from Tuṣita Heaven into [his] mother's womb.' Here [in *Luoyang qielan ji*], it is written: 'A white elephant with six teeth carried Śākyamuni on its back through the void'; this means, the sign of the descent [降生][31] of the Buddha.

Zhou here cites content from *pusa jiangshen pin* 菩薩降神品 of the *Xiuxing benqi jing* 修行本起經, and refers to similar details from *jiangshen chutai pin* 降神處胎品 of the PYJ. They both clearly refer to the Buddha entering into his mother's womb rather than the stage of giving birth. Therefore, if we solely rely on Zhou's commentary we can see that 降生 is to be understood as conception.

However, when Zhou explains the following text from the *Luoyang qielan ji*: "This [kind of] statue usually appears around the 4th day of the 4th lunar month 四月四日此像常出", he cites information related to the birth stage of the Buddha and connects it with Date A (the 8th days of the 4th lunar month). He comments as follows[32]:

案：' 佛於四月八日夜從母右脅而生'。佛既涅槃，後人恨未能親覩真容，故於是日立佛降生相，或太子巡城像······

'The Buddha was born from his mother's flank on the 8th day of the 4th lunar month [Date A]'. After the Buddha reaches *nirvāṇa,* later generations regret not having the chance to see the original face of the Buddha. Therefore, [some people] on this date erected an image of the descent [降生] of the Buddha or [others carried] a statue of the Buddha around the city······

This passage describes how the statue of the birth of the Buddha is usually taken out on the 4th day of the 4th lunar month, even though Zhou adopts Date A (the 8th day of the 4th lunar month) in his commentary. He does not explain the connection between the 4th day of the 4th lunar month and Date A, but stresses that the Buddha comes out from his mother's flank on the night of Date A and people erected an image of the descent of the Buddha for celebrating Date A. In other words, Zhou understands that Date A is related to the Buddha's birth according to this commentary. However, "the descent [降生] of the Buddha" has been indicated in the first commentary above in which 降生 is to be understood as conception, not the birth of the Buddha. Therefore, it is not clear how Zhou arrived at his second interpretation.



In brief, in the two commentaries above, Zhou is inconsistent in his understanding of 降生. In the first case, he takes 降生 to pertain to the conception of the Buddha, although he does not connect it with Date A. In the latter case, he mistakenly uses 降生 to pertain to the birth stage, not the conception, when establishing links with Date A. This fallacy about the conception raises two issues that need to be considered carefully: (1) How to understand the date of the Buddha's conception; (2) Whether the conception occurs on Date A.

Thirdly, both the Indian and Chinese traditions regard an individual's life to begin at the moment of conception. In the Abhidharma of the Sarvāstivāda School, the *Mahāvibhāṣā* 大毗婆沙論 refers to the idea of conception or "entering the mother's womb 入母胎".[33] It suggests "only when three factors work in combination is it possible to enter the mother's womb".[34] Here, the "three factors" mean "the male parent, the female parent, and the *gandharva*".[35] The *gandharva* 健達縛 is the state in which one resides immediately after "death 死有" and before the stage of "(re)birth 生有",[36] but it is also called "intermediate existence 中有 (*antarābhava*)" according to the *Mahāvibhāṣā*.[37] In the continuum of death and rebirth, the "intermediate existence" would start only after the disappearance of "death".[38] And not until "intermediate existence" or "death" has passed, does "(re) birth" begin.[39]

Gotō Toshifumi's 後藤敏文 (Gotō 2009, p. 32) analysis of Vedic literature shows that contexts related to descending from heaven are mostly connected to discussions about entering the womb and the *gandharva* is critical for the ascent and descent. What is more, according to late Vedic sources (3 BCE) regarding the *upanayana*, a person's age is calculated from the date of conception (Kajihara 2003, p. 73, note 11). It is therefore clear that, already from the time of the Vedic literature, the start of someone's life is measured not from the day they are born, but from the date of conception.

Interestingly, this Indian tradition is perfectly in agreement with traditional Chinese ideas. The critical study of Zhang Rongqiang 張榮強 (Zhang 2015, p. 51) reminds us that there are two ways to calculate the age of Chinese people. The way called "Zhousui 周歲" means someone is zero years old when born. The other one is called "Xusui 虛歲", according to which someone will be one year old on the day of birth. Further, Zhang Rongqiang points out that "Xusui" was the mainstream way of counting for historical figures until the 1950s, which indicates that historically, the Chinese also took the moment of conception as the start of life. Therefore, the so-called Buddha's birthday could be argued as his date of conception.

## 4. The 8th Day of the 4th Lunar Month: The Connection to the Month of Vaiśākha

As already seen, both the PYJ and FGJ agree that the month of Vaiśākha, along with some additional information related to the season, is the core of the date of conception. Additionally, I have argued that the Buddha's birthday should be understood as marking the date of conception. Therefore, it is time to ascertain how Vaiśākha corresponds to the Chinese calendar in ancient China.

As Shen Yue 沈約 (441–513) wrote, "The date of the Buddha's birth could not be known. There are no years or dates written in Buddhist *sūtra*s, and the way [of matching the Indian calendar to the Chinese] has not been transmitted into eastern areas yet. Thus, how could we know that [the date of the Buddha's birth] was during the Zhou and Zhuang [of the Spring and Autumn period]"[40] This suggests that few people knew the conversion between Indian and Chinese calendars, or we can say that Indian astrology was not widely studied or popularized in Shen Yue's time.[41] They mostly relied on the record of the *Spring and Autumn Annals* and the *Zuozhuan*, or the *Bamboo Annals* 竹書紀年 and other Chinese chronicles when discussing the date of the Buddha's birth (Liu 2017, p. 72; Chen 2018, p. 121; Franke 1991, p. 443). However, if we do not refer to the Indian calendar, understanding the materials related to the month of Vaiśākha and *puṣyanakṣatra* is not possible.

There are three main texts presenting the Indian astrological elements in Chinese Buddhism. These are Dharmarakṣa's (223–316) translation of the *Śārdūlakarṇāvadāna*,[42]

the chapter on astrology 星宿品 of the *Sūryagarbhaparivarta* 日藏分 of the *Da fangdeng daji jing* 大方等大集經 translated by Narendrayaśas 那連提耶舍 (489–589),[43] and a compilation of astrological lore known by its abbreviated title of *Xiuyao jing* 宿曜經. However, Dharmarakṣa's translation, unlike later Chinese translations, semantically translated Indian *nakṣatra* rather than using the Chinese lunar mansions (xiu 宿). The *Xiuyao jing* is based on non-Buddhist astrology, and there is no known parallel of this work in Sanskrit or Tibetan.[44] Thus I employ Narendrayaśas's translation to discuss the correspondence between Indian and Chinese months (see Table 2).

**Table 2.** Dates corresponding to the Lunar Mansions according to the *Mahāsaṃnipāta*.

| Date 1 [1] | Date 2 | Indian Date [2] | Jan | Feb | Mar | Apr | May | Jun | Jul | Aug [3] | Sep | Oct | Nov | Dec |
|---|---|---|---|---|---|---|---|---|---|---|---|---|---|---|
| 16 January | 16 December | 1st Kṛṣṇapakṣa | 軫 | 亢 | 房 | 尾 | 斗 | 虛 | 辟 | 胃 | 畢 | 參 | 柳 | 張 |
| 17 January | 17 December | 2nd Kṛṣṇapakṣa | 角 | 氐 | 心 | 箕 | 牛、女 | 危 | 奎 | 昴 | 觜 | 井 | 星 | 翼 |
| 18 January | 18 December | 3rd Kṛṣṇapakṣa | 亢 | 房 | 尾 | 斗 | 虛 | 室 | 婁 | 畢 | 參 | 鬼 | 張 | 軫 |
| 19 January | 19 December | 4th Kṛṣṇapakṣa | 氐 | 心 | 箕 | 牛 | 危 | 辟 | 胃 | 觜 | 井 | 柳 | 翼 | 角 |
| 20 January | 20 December | 5th Kṛṣṇapakṣa | 房 | 尾 | 斗 | 女 | 室 | 奎 | 昴 | 參 | 鬼 | 星 | 軫 | 亢 |
| 21 January | 21 December | 6th Kṛṣṇapakṣa | 心 | 箕 | 牛 | 虛 | 辟 | 婁 | 畢 | 井 | 柳 | 張 | 角 | 氐 |
| 22 January | 22 December | 7th Kṛṣṇapakṣa | 尾 | 斗 | 女 | 危 | 奎 | 胃 | 觜 | 鬼 | 星 | 翼 | 亢 | 房 |
| 23 January | 23 December | 8th Kṛṣṇapakṣa | 箕 | 牛 | 虛 | 室 | 婁 | 昴 | 參 | 柳 | 張 | 軫 | 氐 | 心 |
| 24 January | 24 December | 9th Kṛṣṇapakṣa | 斗 | 女 | 危 | 辟 | 胃 | 畢 | 井 | 星 | 翼 | 角 | 房 | 尾 |
| 25 January | 25 December | 10th Kṛṣṇapakṣa | 牛 | 虛 | 室 | 奎 | 昴 | 觜 | 鬼 | 張 | 軫 | 亢 | 心 | 箕 |
| 26 January | 26 December | 11th Kṛṣṇapakṣa | 女 | 危 | 辟 | 婁 | 畢 | 參 | 柳 | 翼 | 角 | 氐 | 尾 | 斗 |
| 27 January | 27 December | 12th Kṛṣṇapakṣa | 虛 | 室 | 奎 | 胃 | 觜 | 井 | 星 | 軫 | 亢 | 房 | 箕 | 牛 |
| 28 January | 28 December | 13th Kṛṣṇapakṣa | 危 | 辟 | 婁 | 昴 | 參 | 鬼 | 張 | 角 | 氐 | 心 | 斗 | 女 |
| 29 January | 29 December | 14th Kṛṣṇapakṣa | 室 | 奎 | 胃 | 畢 | 井 | 柳 | 翼 | 亢 | 房 | 尾 | 牛 | 虛 |
| 30 January | 30 December | 15th Kṛṣṇapakṣa | 辟 | 婁 | 昴 | 觜 | 鬼 | 星 | 軫 | 氐 | 心 | 箕 | 女 | 危 |
| 1 February | 1 January | 1st Śuklapakṣa | 奎 | 胃 | 畢 | 參 | 柳 | 張 | 角 | 房 | 尾 | 斗 | 虛 | 室 |
| 2 February | 2 January | 2nd Śuklapakṣa | 婁 | 昴 | 觜 | 井 | 星 | 翼 | 亢 | 心 | 箕 | 牛 | 危 | 辟 |
| 3 February | 3 January | 3th Śuklapakṣa | 胃 | 畢 | 參 | 鬼 | 張 | 軫 | 氐 | 尾 | 斗 | 女、虛 | 室 | 奎 |

**Table 2.** *Cont.*

| Date 1 [1] | Date 2 | Indian Date [2] | Jan | Feb | Mar | Apr | May | Jun | Jul | Aug [3] | Sep | Oct | Nov | Dec |
|---|---|---|---|---|---|---|---|---|---|---|---|---|---|---|
| 4 February | 4 January | 4th Śuklapakṣa | 昴 | 觜 | 井 | 柳 | 翼 | 角 | 房 | 箕 | 牛 | 危 | 辟 | 婁 |
| 5 February | 5 January | 5th Śuklapakṣa | 畢 | 參 | 鬼 | 星 | 軫 | 亢 | 心 | 斗 | 女 | 室 | 奎 | 胃 |
| 6 February | 6 January | 6th Śuklapakṣa | 觜 | 井 | 柳 | 張 | 角 | 氐 | 尾 | 牛 | 虛 | 辟 | 婁 | 昴 |
| 7 February | 7 January | 7th Śuklapakṣa | 參 | 鬼 | 星 | 翼 | 亢 | 房 | 箕 | 女 | 危 | 奎 | 胃 | 畢 |
| 8 February | 8 January | 8th Śuklapakṣa | 井 | 柳 | 張 | 軫 | 氐 | 心 | 斗 | 虛 | 室 | 婁 | 昴 | 觜 |
| 9 February | 9 January | 9th Śuklapakṣa | 鬼 | 星 | 翼 | 角 | 房 | 尾 | 牛、女 | 危 | 辟 | 胃 | 畢 | 參 |
| 10 February | 10 January | 10th Śuklapakṣa | 柳 | 張 | 軫 | 亢 | 心 | 箕 | 虛 | 室 | 奎 | 昴 | 觜 | 井 |
| 11 February | 11 January | 11th Śuklapakṣa | 星 | 翼 | 角 | 氐 | 尾 | 斗 | 危 | 辟 | 婁 | 畢 | 參 | 鬼 |
| 12 February | 12 January | 12th Śuklapakṣa | 張 | 軫 | 亢 | 房 | 箕 | 牛、女 | 室 | 奎 | 胃 | 觜 | 井 | 柳 |
| 13 February | 13 January | 13th Śuklapakṣa | 翼 | 角 | 氐 | 心 | 斗 | 虛 | 辟 | 婁 | 昴 | 參 | 鬼 | 星 |
| 14 February | 14 January | 14th Śuklapakṣa | 軫 | 亢 | 房 | 尾 | 牛 | 危 | 奎 | 胃 | 畢 | 井 | 柳 | 張 |
| 15 February | 15 January | 15th Śuklapakṣa | 角 | 氐 | 心 | 箕 | 女 | 室 | 婁 | 昴 | 觜 | 鬼 | 星 | 翼 |

[1] I add the columns "Date 1"and "Date 2" for the Chinese calendar. The additional data are based on the *Da fangdeng daji jing*, except the column of "Aug". [2] For the sake of brevity, "day of the" inside "1st [day of the] Kṛṣṇapakṣa" is omitted in the column "Indian Date". [3] Narendrayaśas does not provide the *nakṣatra* dates of "Aug", this column is calculated by me according to the preceding and succeeding *nakṣatra* dates.

Indian months were divided into fortnights called *kṛṣṇapakṣa* and *śuklapakṣa* as reported in the *Great Tang Records on the Western Regions* 大唐西域記 narrated by Xuanzang 玄奘 (602–664). The *śuklapakṣa* is from the new moon day to the full moon day, and the *kṛṣṇapakṣa* is from the full moon day to the day before the new moon. A whole month starts with *kṛṣṇapakṣa* and is followed by *śuklapakṣa*.[45] Xuanzang provided his rendition for each Indian month and its periods (see Table 3).

As can be seen in Table 2, the 15th day of the *śuklapakṣa* of the first Indian month corresponds to the constellation Jiaoxiu 角宿. There are two possibilities when matching the numerical dates in China with their counterparts in India. For example, 16 January corresponds to the 1st day of the *kṛṣṇapakṣa*, and then the 15th day of the *śuklapakṣa* corresponds to 15 February. According to this system, the month of Vaiśākha (Dixiu) is from 16 February to 15 March, which is exactly what Xuanzang shows. The other possibility is that December 16th corresponds to the 1st day of the *kṛṣṇapakṣa*, and January 15th corresponds to the 15th day of the *śuklapakṣa*. Then the month of Vaiśākha is from 16 January to 15 February.

**Table 3.** The Indian 27 Nakṣatra and months according to Xuanzang 玄奘.

| Nakṣatra [1] | Chinese Name of Nakṣatra [2] | Chinese Names of Month | Transliteration of Month [3] | Month | The Period of Month (According to Xuanzang) |
|---|---|---|---|---|---|
| Aśvayujau | lou 娄 | 娄月 / 八月 August [4] | 頞濕縛庾闍月 | Āśvina | from July 16 to August 15 |
| Bharaṇyaḥ | wei 胃 | | | | |
| Kṛttikāḥ | mao 昴 | 昴月 / 九月 September | 迦剌底迦月 | Kārttika | from August 16 to September 15 |
| Rohinī | bi 畢 | | | | |
| Mṛgaśiras | zi 觜 | 觜月 / 十月 October | 末伽始羅月 | Mārgaśira | from September 16 to October 15 |
| Ārdrā | shen 參 | | | | |
| Punarvasū | jing 井 | | | | |
| Puṣya | gui 鬼 | 鬼月 / 十一月 November | 報沙月 | Pauṣa | from October 16 to November 15 |
| Āśleṣāḥ | liu 柳 | | | | |
| Maghāḥ | xing 星 | 星月 / 十二月 December | 磨祛月 | Māgha | from November 16 to December 15 |
| Pūrva-phalgunī | zhang 張 | | | | |
| Uttara-phalgunī | yi 翼 | 翼月 / 一月 January | 頗勒窶拏月 | Phālguna | from December 16 to January 15 |
| Hasta | zhen 軫 | | | | |
| Citrā | jiao 角 | 角月 / 二月 February | 制呾羅月 | Caitra | from January 16 to February 15 |
| Svāti | kang 亢 | | | | |
| Viśākhe | di 氐 | 氐月 / 三月 March | 吠舍佉月 | Vaiśākha | from February 16 to March 15 |
| Anurādhā | fang 房 | | | | |
| Jyeṣṭhā | xin 心 | 心月 / 四月 April | 逝瑟吒月 | Jyaiṣṭha | from March 16 to April 15 |
| Nūla | wei 尾 | | | | |
| Pūrvāṣāḍhāḥ | ji 箕 | 箕月 / 五月 May | 頞沙荼月 | Āṣāḍha | from April 16 to May 15 |
| Uttarāṣāḍhāḥ | dou 斗 | | | | |
| Śravaṇa | nü 女 | 女月 / 六月 June | 室羅伐拏月 | Śrāvaṇa | from May 16 to June 15 |
| Dhaniṣṭhāḥ | xu 虛 | | | | |
| Śatabhiṣa | wei 危 | | | | |
| Pūrva-proṣṭhapadāḥ | shi 室 | 室月 / 七月 July | 婆羅鉢陀月 | Bhādrapada | From June 16 to July 15 |
| Uttara-bhadrapadadāḥ | bi 壁 | | | | |
| Revatī | kui 奎 | | | | |

[1] For the Sanskrit names of the *nakṣatra* and months, see Renou (Renou 1979–1981, pp. 363–64), and Yano (1992, p. 95, Table 2). [2] See the *Wenshu shili pusa ji zhuxian suoshuo jixiong shiri shane xiuyao jing*, T no. 1299, 21: 394c20–23, and Yano (2013, p. 76). [3] To compare with Table 2, this column is based on the *Datang xiyuji* 大唐西域記. [4] For the sake of brevity, I use the western month names, e.g., "January", "February", to indicate corresponding Chinese terms "yiyue 一月", "eryue 二月". However, it should be noted that the Xia, Shang and Zhou calendars of China adopt different months as the first month of the year.

These two methods of converting the month of Vaiśākha to the Chinese calendar can also be noticed in the Chinese translations of Pāli texts. For example, the time of the Buddha's nirvāṇa suggested in the Samantapāsādikā is *visākha* (the month's name) *puṇṇama* (full moon) *divasa* (day) *paccūsasamaya* (dawn moment) (Takakusu 2008, p. 4). The Chinese translation by Saṁghabhadra 僧伽跋陀羅 (5CE?) is "the dawn of February 15th".[46] There is *visākha* (the month's name) *puṇṇamāya* (full moon) *hi'ssa* (his) *abhisekaṃ* (consecration) *akaṃsu* (to be) in the same text (Takakusu 2008, p. 76), but its Chinese translation is suggested as "at March 15th, [Devānajpiyatissa] accepts his consecration".[47] The difference of February 15th and March 15th in Chinese translations here is strong evidence for the above two possibilities.

Another issue is the divergence between the views of Mahāyāna and Theravāda Buddhists on the Buddha's birthday. Xuanzang pointed out that "[the date of the Buddha's birth is on] the 8th day of *śuklapakṣa* in the month of Vaiśākha which corresponds to the 8th of March. [But] Theravāda [Buddhism] holds that it is on the 15th *śuklapakṣa* in the month of Vaiśākha and it corresponds to the 15th of March".[48] This also leads to the question whether the 8th day of the *śuklapakṣa* in the month of Vaiśākha was considered as Date A or Date B.

Dharmarakṣa's translation of the *Śārdūlakarṇāvadāna* tells us that "On the 8th day of the 12th lunar month during winter, there are 18 *muhūrta* 須臾 in the night and 12 *muhūrta* in the day. On Date A during the spring, there are 18 *muhūrta* in the day and 12 in the night."[49] Another text that describes the length of day and night in India, the *Jushelun shu* 俱舍論疏 by Fabao 法寶 (625?–733?), a disciple of Xuanzang, is a commentary on the *Abhidharmakośa*. He mostly refers to the *Abhidharma Mahāvibhāṣā* in his commentary.

Because India and China are both in the Northern Hemisphere, their vernal and autumnal equinoxes should be consistent. According to the expression in the *Abhidharma Mahāvibhāṣā* "on the 8th day of the *śuklapakṣa* in the month of Kārttika, there are equal 15 *muhūrta* both in the day and in the night",[50] we know that the 8th day of the *śuklapakṣa* in the month Kārttika should be the vernal or autumnal equinox. Moreover, "after that (the 8th day of the *śuklapakṣa* in the month Kārttika), daytime will decrease one *lava* and the time of night will increase one *lava*".[51] This indicates that the 8th day of the *śuklapakṣa* in the month Kārttika must be the autumnal equinox. In a similar way, we know that the 8th day of the *śuklapakṣa* in the month of Vaiśākha is the vernal equinox in India (see Table 4).

Significantly, all equinoxes in India are recorded as the 8th *śuklapakṣa* of a month, which strongly implies that the date of conception, the 8th *śuklapakṣa* of the month Vaiśākha, should be one of the equinoxes. The time references in the Chinese translations of the *Lalitavistara* show that the climate during the month Vaiśākha should be "neither cold nor hot".[52] This is the exact climate of the vernal equinox recorded in Chinese sources as well. The *Luxuriant Dew of the Spring and Autumn Annals* 春秋繁露 states that "The vernal equinox, is half *ying* and half *yang*. Therefore, the day and night are equal to each other and there is an even balance of cold and heat".[53] Thus, it is assured that the date of conception of the Buddha is the vernal equinox in China as well.

Careful comparison of Tables 3 and 4 reveals that the correspondence of the Indian and Chinese months is different. The *Jushelun shu* vividly describes Fabao's consideration, which is different from Xuanzang's.[54] Elsewhere, Fabao pointed out that Xuanzang included additional text in the Chinese translation of the *Abhidharma Mahāvibhāṣā* to facilitate understanding;[55] this point suggests Fabao may have had reason to consider Xuanzang less reliable and thus he proposed an alternative interpretation, without reference to Xuanzang, of the correspondence between the Indian and Chinese months.

**Table 4.** The duration of the Indian day and night, and Chinese solar terms according to Fabao.

| Transliteration of Month | Month | Date (*Mahāvibhāṣāśāstra*) | Duration of the Day and Night | Solar Terms | Date (*Jushelunshu*) | Period of the Month (*Jushelunshu*) |
|---|---|---|---|---|---|---|
| 羯栗底迦月 | Kārttika | 白半第八日<br>The 8th day of the *śuklapakṣa* | 晝夜各十五<br>day and night are both fifteen (lavas) | 秋分<br>autumnal equinox | 八月八日<br>August 8th | from July 16 to August 15 |
| 末伽始羅月 | Mārgaśira | 白半第八日<br>The 8th day of the *śuklapakṣa* | 夜十六晝十四<br>night is fourteen day is sixteen (lavas) | | | from August 16 to September15 |
| 報沙月 | Pauṣa | 白半第八日<br>The 8th day of the *śuklapakṣa* | 夜十七晝十三<br>night is seventeen day is thirteen (lavas) | | | from September 16 to October 15 |
| 磨伽月 | Māgha | 白半第八日<br>The 8th day of the *śuklapakṣa* | 夜十八晝十二<br>night is eighteen day is twelve (lavas) | 冬至<br>winter solstice | 十一月八日<br>December 8th | from October 16 to December 15 |
| 頗勒寠那月 | Phālguna | 白半第八日<br>The 8th day of the *śuklapakṣa* | 夜十七晝十三<br>night is seventeen day is thirteen (lavas) | | | from December 16 to November 15 |
| 制怛羅月 | Caitra | 白半第八日<br>The 8th day of the *śuklapakṣa* | 夜十六晝十四<br>night is fourteen day is sixteen (lavas) | | | from November 16 to January 15 |
| 吠舍佉月 | Vaiśākha | 白半第八日<br>The 8th day of the *śuklapakṣa* | 晝夜各十五<br>day and night are both fifteen (lavas) | 春分<br>spring equinox | 二月八日<br>February 8th | from January 16 to February 15 |
| 誓瑟搋月 | Jyaiyaṣṭha | 白半第八日<br>The 8th day of the *śuklapakṣa* | 夜十四晝十六<br>night is fourteen day is sixteen (lavas) | | | from February 16 to March 15 |
| 阿沙荼月 | Āṣāḍha | 白半第八日<br>The 8th day of the *śuklapakṣa* | 夜十三晝十七<br>night is thirteen day is seventeen (lavas) | | | from March 16 to April 15 |
| 室羅筏拏月 | Śrāvaṇa | 白半第八日<br>The 8th day of the *śuklapakṣa* | 夜十二晝十八<br>night is twelve day is eighteen (lavas) | 夏至<br>summer solstice | 五月八日<br>May 8th | from April 16 to May 15 |
| 婆達羅缽陀月 | Bhādrapada | 白半第八日<br>The 8th day of the *śuklapakṣa* | 夜十三晝十七<br>night is thirteen day is seventeen | | | from May 16 to June 15 |
| 阿濕縛庾闍月 | Āśvina | 白半第八日<br>The 8th day of the *śuklapakṣa* | 夜十四晝十六<br>night is fourteen day is sixteen | | | from June 16 toJuly 15 |

　　There is evidence that Fabao was active during the period of Wu Zetian's 武則天 rule (r.690–705 CE).[56] At the beginning of the *Jushelun shu*, Fabao refers to different calendars of India and the Tang Dynasty: "India takes the first month (*zi* 子) of the earthly branches as the first month, but here, previously China employed the third month (*yin* 寅) of the earthly branches as the first month".[57] Afterwards, he explains "the 8th day of *śuklapakṣa* in

the month of Vaiśākha (here, it should be the 8th day of the 2nd lunar month [Date B])".[58]
If we pay attention to "here, previously 此方, 先時" and "here 此方", we will notice that at
the time when he wrote this work, the calendar in use differed from the "previous" one.
In fact, Wu Zetian reformed the calendar during her rule. The *Ritual Annal of Jiu Tang shu*
舊唐書·禮儀誌 states that "the first year of Zaichu 載初 (690), on the Gengchen 庚辰 day,
the first day of the first month, an amnesty was declared and from then, the Zhou calendar
was applied".[59] Then, in the tenth month of the first year of Jiushi 久視 (700) "there was a
return to the previous calendar, *Yiyue* 一月 was changed to *Zhengyue* 正月 and it was the
beginning of the year again".[60] Therefore, "the 8th day of the 2nd lunar month [Date B]"
should be understood according to the Zhou calendar of the period of Wu Zetian's rule.
Nevertheless, "*eryue* 二月" of the Wuzhou calendar, based on Wu Zetian's reforms, is the
4th lunar month of the Xia calendar (see Table 5).

**Table 5.** The calendars of Zhou, Xia and Wuzhou.

| Order of Month | 1st | 2nd | 3rd | 4th | 5th | 6th | 7th | 8th | 9th | 10th | 11th | 12th |
|---|---|---|---|---|---|---|---|---|---|---|---|---|
| Xia | 正月 January | 二月 February | 三月 March | 四月 April | 五月 May | 六月 June | 七月 July | 八月 August | 九月 September | 十月 October | 十一月 November | 十二月 December |
| Zhou | 正月 November | 十二月 December | 一月 January | 二月 February | 三月 March | 四月 April | 五月 May | 六月 June | 七月 July | 八月 August | 九月 September | 十月 October |
| Wuzhou | 正月 November | 臘月 December | 一月 January | 二月 February | 三月 March | 四月 April | 五月 May | 六月 June | 七月 July | 八月 August | 九月 September | 十月 October |

So far, I have verified the date of conception of the Buddha, the 8th day of the *śuklapakṣa*
of the month Vaiśākha, as the vernal equinox. It is Date B according to the Wuzhou cal-
endar, but it is also Date A in the Xia calendar. Accordingly, after ten months' pregnancy,
the date of delivery of the Buddha is Date B in the Xia calendar.

Hence, the confusion about Date A or Date B for the four main events of the Buddha's
life, especially for his birthday, should probably be traced to the failure to notice the dif-
ference between the date of conception and the date of birth. Early Chinese translations
of the Buddha's biographical literature were not translated during the Zhou or Wuzhou
dynasties and therefore did not use those dynasties' calendars; rather, Date A and Date B
were both converted to the Xia calendar and respectively indicate the date of conception
and the date of birth of the Buddha.

## 5. Concluding Remarks

The Buddha's birthday is celebrated on various dates across Asia, even today. Al-
though Japan has adopted the Gregorian calendar, and consequently the birth of the Bud-
dha is celebrated there on a different date from other countries in Asia, it is universally
admitted that Date A (the 8th day of the 4th lunar month), corresponding to 8 April in
modern Japan, derives from the Chinese Buddhist literature. It is, however, not easy to
solve the contradiction caused by both Date A and Date B (the 8th day of the 2nd lunar
month) being taken as the Buddha's birthday in some texts in Chinese Buddhism.

I proposed a rather new interpretation to try to solve this puzzle, and provide an
answer to the question, "whence the 8th day of the 4th lunar month?". My suggestion
may be summarized in the following three points:

1. In addition to the date indicated by month (月) and day (日), there is an auspicious
junction star, star Fei, which is also an indispensable element for establishing the Buddha's
birthday. The date of conception and the date of birth of the Buddha, both of which refer
to this auspicious junction star, were variously translated into both Date A and Date B in
early Chinese Buddhist literature.

2. Of the three stages of the Buddha's birth, his conception contains the most detailed
time expressions, in particular reference to the month Vaiśākha. The fact that, in India,
all the main events in the Buddha's life are celebrated on the same day in Vaiśākha likely

derives from the date of conception, which is understood as the beginning of the Buddha's present life.

3. The date of conception of the Buddha could be specified as the 8th day of the *śuklapakṣa* of the month Vaiśākha, the vernal equinox in India and China. Its Chinese date is Date A in the Xia calendar. Date B is the date of birth of the Buddha after ten months' pregnancy.

**Funding:** This research received no external funding.

**Institutional Review Board Statement:** Not applicable.

**Informed Consent Statement:** Not applicable.

**Data Availability Statement:** Not applicable.

**Acknowledgments:** I am grateful to the following professors who shared with me their knowledge about the studies and materials related to the Indian calendars: Hori Shin'ichiro 堀伸一郎, Gotō Toshifumi's 後藤敏文 and Gotō (Sakamoto) Junko 後藤（坂本）純子. I would also like to express my deep gratitude to Hao Chunwen, Michael Zimmermann, Florin Deleanu and Gotō Toshifumi who read my very early drafts of this paper and gave me their incisive comments. The present version has been changed quite a lot during these years' learning and revisions. I am deeply indebted to Jonathan Silk and Corin Golding who polished my English and generously shared their useful comments. All errors are mine.

**Conflicts of Interest:** The author declares no conflict of interest.

## Abbreviations

| | |
|---|---|
| T | *Taishō shinshū daizōkyō* 大正新脩大藏經. Ed. Takakusu Junjirō 高楠順次郎 and Watanabe Kaikyoku 渡辺海旭. Tokyo: Taoshō Issaikyō Kankōkai. 1924–1932. |
| JTS | *Jiutangshu* 舊唐書. Ed. Liu Xu 劉昫. Beijing: Zhonghua shuju 中華書局. 1975. |
| DN | *The Dīgha Nikāya.* Eds. T.W. Rhys Davids and J. Estlin Carpenter. London: Pali Text Society. 1996. |

## Notes

1.  Most places in China and Korea celebrate on April 8 under the Asian lunisolar calendar, while the Japanese have celebrated on April 8 under the Gregorian calendar since the Meiji era. See Gorai (2009).
2.  夏，恒星不見，夜明也. *Chunqiu jingzhuan jijie* 春秋經傳集解, p. 142.
3.  到四月八日夜，明星出時，化從右脇生. (T185: 473c).
4.  The end of the *Mahāparinibbāṇa Sutta* of the *Digha Nikaya* extends only up to the disputation on the Buddha's śarīra among eight kings. See DN: pp. 167–68.
5.  何等生二足尊？何等出叢林苦？何等得最上道？何等入涅槃城？沸星生二足尊 … … 八日如來生 … … 二月如來生. (T1: 30a).
6.  佛以四月八日生，八日棄國，八日得道，八日滅度，以沸星時，去家學道，以沸星時得道，以沸星時般泥. (T5: 175c). 佛從四月八日生，四月八日捨家出，四月八日得佛道，四月八日般泥洹，皆以* 佛星出時. (T6: 190c). *The Second Edition of the Korean Canon has 佛, while the Three editions have 沸.
7.  "Here" is used in a general sense to refer to various places in Ancient China, to contrast with or distinguish from the text's Indic origins.
8.  A near-parallel passage can be found in the commentary by Chengguan 澄觀 (738–839, who was known as Master Qingliang 清涼國師), on the Avataṃsaka Sūtra [*Da fangguangfo huayan jing*] 大方廣佛華嚴經, the *Da fangguangfo huayan jingshu* 大方廣佛華嚴經疏: "Fusha is also named 'Bo sha'. Here, it means 'prosper'. Because the meaning of insight and ultimate truth is 'prosper'. 弗沙亦云勃沙. 此云增盛. 明達勝義是增盛也" (T1735: 628c).
9.  Tiṣya and puṣya (the star Fei's Sanskrit counterpart) are two Buddha's names as well. See Tournier (2017, pp. 158–60).
10. 弗沙, 正名富沙. 清涼[大師] 云: "亦云勃沙. 此云增盛, 明達勝義故也". 亦云底沙, 亦云提舍. 此翻明, 又云說度, 說法度人也. 什師解弗沙菩薩云: "二十八宿中鬼星名也. 生時相應鬼宿, 因以為名. 或名沸星, 或名孛星". (T2131: 1058c).
11. 菩薩不於黑月入胎，要以白月弗沙星合. (T187: 543a).
12. Windisch (1908, p. 159). I express my gratitude to Professor Gotō Toshifumi 後藤敏文 for this reference. There is no study about the relationship between the *Mahāvastu* and the *Lalitavistara*, but there are some identical sentences which they share; see Tournier (2017, pp. 118–19).

13 " 據白月十五日夜太陰所在宿為月名", see T1299: 388a.

14 According to Xuanzang and Fabao (see below), the month of Pauṣa does not correspond to December–January but rather to October–November and September–October (see Tables 3 and 4); ten months later is July–August and June–July. They are still not close to Date A or Date B.

15 弗沙菩薩曰, 什曰:" 二十八宿中鬼星名也. 生時所值宿, 因以為名也". 肇曰: " 弗沙星名也, 菩薩因以為字焉". (T1775: 397b).

16 菩薩欲入母胎之時, 取鬼宿日, 然後乃入於母胎中. (T190: 679c).

17 Importantly, a connection between the the *Foben xingji jing* and the *Mahāvastu* has been identified. See Mizuno (1964).

18 用沸星吉日. (T1435: 90b28).

19 鬼宿主於一切國王大臣. (T397: 371a).

20 諸宿中, 鬼宿為最. (T893: 625c).

21 牟尼今者, 富沙 (唐言鬼宿) 星生. (T402: 555c). I express my gratitude to Dr. Liu Chang for this reference.

22 See note No.8 of the T402: 555.

23 以四月八日明星出時, 降神母胎. (T189: p. 624a).

24 This difference might suggest that the four events in the Buddha's life sharing the same date, with the same auspicious star, may all derive from references to the date of conception of the Buddha in the early tradition.

25 毘舍佉, 或鼻奢佉. 此云別枝, 即是氐宿. (T1231: 1086a).

26 從初入胎及以出胎, 童子盛年, 遊戲受欲, 出家苦行, 詣菩提座, 降伏魔軍, 轉正法輪現大神力, 下忉利天入般涅槃 (T187: 546b–c). For its Sanskrit, see Hokazono (1994, p. 356). Its other Chinese translation renders as " 降神入胎不離其側, 如影隨形, 乃至成佛, 降伏魔官, 而轉法輪, 和慈四等至大滅度". (T186: 489a).

27 The most detailed description of conception among the Buddhist texts is in the *Garbhāvākrāntisūtra*, in one of its Chinese translations titled *Baotai jing* 胞胎經 T317. For studies, see Kritzer (2014), and Langenberg (2017). However, neither of these notice this point. Furthermore, Kritzer does mention the process of rebirth, but as he says, rebirth begins at the moment of death in one life and the moment of conception is no more than one step of rebirth. See Kritzer (2009, p. 73). However, this is different from taking conception as the beginning of one's life.

28 These three parts are according to the *Fang guang dazhuangyan jing* 方廣大莊嚴經. (T187: 545–57).

29 Neither *jiangsheng* nor *jiangshen* have their Sanskrit correspondence in the *Lalitavistara*. See Kawano (2007, pp. 195–98).

30 *Luoyang qielan ji* 洛陽伽藍記: 36.

31 It is not possible to state with certainty how Zhou understood this term here. The English "descent" retains the ambiguity in *jiangshen*, which as we have seen is interpreted as both conception and birth.

32 See Note 30 above.

33 See T1545: 363b.

34 三事和合, 得入母胎. T1545: 363a.

35 父及母并健達縛三事和合. T1545: 363b.

36 See T1545: 363a.

37 或名中有, 或名捷達縛. (T1545: 363a). However, originally, *antarābhava* 中有 was not the same as *gandharva* 健達縛. The former is an intermediate state between death and the next life during *saṃsāra*, while the latter has a very ancient origin which could be traced back to the *Rigveda* and has been developed with different meanings in later times. See Ogawa (1990, pp. 106–8).

38 中有相續者, 謂死有蘊滅, 中有蘊生. (T1545: 310a).

39 生有相續者, 謂中有蘊滅, 或死有蘊滅, 生有蘊生. (T1545: 310a).

40 釋迦出世年月, 不可得知. 佛經既無年曆注記, 此法又未東流, 何以得知是周莊之時. (T2103: 122b).

41 Kotyk argues that "astrology as an essential practice within a Buddhist framework only became popular after the introduction of Mantrayāna", see Kotyk (2017, p. 55).

42 It is titled as *Shetoujian taizi ershibaxiu jing* 舍頭諫太子二十八宿經, or *Huer jing* 虎耳經. For the study of its Sanskrit, see Kotyk (2017); Zhou (2020).

43 T397: 280b–282a.

44 Zenba (1968, pp. 3–6). In addtion, the recension in the Taishō is not the original version of this work; the earlier version has been stored in Japan in manuscripts and printed books. See (Yano 2013, pp. 148–51); Yano (2016, pp. 6–10).

45 月盈至滿, 謂之白分. 月虧至晦, 謂之黑分 ... ... 黑前白後, 合為一月. Ji (2000, p. 168).

46 二月十五平旦. (T1462: 673b).

47 以三月十五日受拜王位. (T1462: 688a).

48 吠舍佉月後半八日, 當此三月八日. 上座部則曰以吠舍佉月後半十五日, 當此三月十五日. Ji (2000, p. 523).

49 冬時十二月八日, 夜有十八須臾. 春四月八日, 晝日有十八須臾耳, 夜有十二須臾. (T1301: 416b). Monier-Williams (1899), s.v. *muhūrta*: "a moment, instant, any short space of time [ ... ]; a partic. division of time, the 30th part of a day, a period of 48 min."

[50] 羯栗底迦月白半第八日, 晝夜各十五牟呼栗多. (T1545: 701c).

[51] 從此以後，晝減夜增各一臘縛. (T1545: 701c).

[52] 不寒不暑. (T186: 491a), 不寒不熱 (T187: 548c).

[53] 春分者，陰陽相半也，故晝夜均而寒暑平 (*The Luxuriant Dew of the Spring and Autumn Annals* 春秋繁露, p. 343).

[54] For details, see T1822: 617–618.

[55] For details, see T2061: 727a.

[56] Fabao attended Śikṣānanda's 實叉難陀 and Yijing's 義淨 translation teams played important roles during Wu Zetian's rule (Wu Zetian proclaimed herself as the emperor from 690 CE, but she ruled in the name of her sons from Gaozong's 高宗 death in 683). See T2074: 176b and T2061: 727b.

[57] 婆羅門國以建子立正, 此方先時以建寅立正. (T1822: 453a).

[58] 吠舍佉月白半第八日（此當此方二月八日）. (T1822: 617b).

[59] 載初元年正月庚辰朔 . . . . . . 大赦改元，用周正. (JTS, 22: 864).

[60] 復舊正朔，改一月為正月，仍以為歲首. (JTS, 6: 129).

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
