# Peer review of "Whence the 8th Day of the 4th Lunar Month as the Buddha’s Birthday"

_religions, doi:10.3390/rel14040451_

Round 1

Reviewer 1 Report

See my comments attached.

Author Response

Please note the lines have been changed automatically in the revised version.

Reviewer 2 Report

Your paper is well written and presents its arguments convincingly in favour of distinguishing the date of the Buddha’s birth from that of his conception. This is a convincing attempt to solve the problem of the varying days accepted by the tradition for the Buddha’s birthday.

Please address the following points:

It is unhelpful to have your endnotes numbered with Roman numerals. Please change to Arabic numerals.

l.52: add ‘not’

ll.139-140: The Skt. seems incomplete when compared to the English translation just provided.

l.223: delete ‘of the Buddha’

l.235: nakṣatra >> add comma

l.306: Shakyamuni >> Śākyamuni

l.376: date of conception of the Buddha >> his date of conception (remove ‘of the Buddha’)

l.437: Does Xuanzang really use the term ‘Theravāda’?

l.472: understanding, >> understanding;

l.523: Vaiśākha >> add comma

Author Response

(The authors gave the same response as above.)
